# Adversarial Attacks on Spiking Convolutional Networks for Event-based Vision

## Abstract

Event-based sensing using dynamic vision sensors is gaining traction in low-power vision applications. Spiking neural networks work well with the sparse nature of event-based data and suit deployment on low-power neuromorphic hardware. Being a nascent field, the sensitivity of spiking neural networks to potentially malicious adversarial attacks has received very little attention so far. In this work, we show how white-box adversarial attack algorithms can be adapted to the discrete and sparse nature of event-based visual data, and to the continuous-time setting of spiking neural networks. We test our methods on the N-MNIST and IBM Gestures neuromorphic vision datasets and show adversarial perturbations achieve a high success rate, by injecting a relatively small number of appropriately placed events. We also verify, for the first time, the effectiveness of these perturbations directly on neuromorphic hardware. Finally, we discuss the properties of the resulting perturbations and possible future directions.

## 1 Introduction

Unlike the usual neural networks of contemporary deep learning, spiking neural networks (SNN) resemble the animal brain more closely in at least two main aspects: the way their neurons communicate through impulses (spikes), and their dynamics, which evolve in continuous time. Aside from offering the field of computational neuroscience more biologically plausible neuron models and communication schemes, research in the technological applications of spiking neural networks is currently blooming because of the rise of neuromorphic technology. Neuromorphic hardware is directly compatible with spiking neural networks and enables the design of low-power models for use in battery-operated, always-on devices.

Adversarial examples are an "intriguing property of neural networks" (Szegedy et al., 2013) by which the network is easily fooled into misclassifying an input which has been altered in an almost imperceptible way by the attacker. This property is usually undesirable in applications: it was proven, for example, that an adversarial attack may pose a threat to self-driving cars, by making them misclassify a stop sign as a speed limit sign; and that this attack can be implemented in the real world through stickers physically placed on the road sign (Eykholt et al., 2018). Because of their relevance to real-world applications, a large amount of work has been published on this subject, typically following a pattern where new attacks are discovered, followed by new defense strategies, in turn followed by proof of other strategies that can still break through them (see Akhtar & Mian (2018) for a review).

With the advent of real-world applications of spiking networks in neuromorphic devices, it is essential to make sure they work securely and reliably in a variety of contexts. In particular, there is a significant need for research on the possibility of adversarial attacks on spiking network models used for computer sensing tasks. In this paper, we make an attempt at modifying event-based data, by adding and removing events, to generate adversarial examples that fool spiking networks into misclassifying them. This offers important insight into the reliability and security of neuromorphic vision devices, with important implications for commercial applications.

## 1.1 WHAT IS EVENT-BASED SENSING?

Event-based cameras, usually called Dynamic Vision Sensors (DVS), share many characteristics with the mammalian retina, which make them excel in some circumstances where traditional frame-based cameras do not perform well (Liu & Delbruck, 2010; Liu et al., 2019b). First, events are generated only when there are changes in the visual scene, automatically removing redundancies; second, their pixels fire independently of each other which means that there is no frame rate, but rather a continuous stream of asynchronous events, so that the latency can be extremely small; third, they have a very high dynamic range which makes them suitable to detect motion in both bright and dark settings. For these reasons, they have found applications in human-robot interaction, odometry, drone control, tracking, and surveillance, including on devices that are already commercially available (Gallego et al., 2019; Kueng et al., 2016; Falanga et al., 2020). Beyond computer vision, the realm of event-based sensing extends to auditory sensors known as silicon cochleas (Chan et al., 2007), as well as radar (Stuijt et al., 2021) and tactile sensors (Caviglia et al., 2016).

Neuromorphic sensors make available a new kind of sparse, asynchronous data, which does not suit current high-throughput, synchronous accelerators such as GPUs. To process event-based data efficiently, a new generation of neuromorphic hardware is being developed in parallel to the spiking neural network models that can be trained in software. Spiking neuromorphic implementations include large-scale simulation of neuronal networks for neuroscience research (Furber et al., 2012) and low-power real-world deployments of machine learning algorithms. In particular, convolutional neural network (CNN) architectures, used for computer vision, have been run on neuromorphic chips such as IBM's TrueNorth (Esser et al., 2016), Intel's Loihi (Davies et al., 2018) and SynSense's Speck and Dynap-CNN hardware (Liu et al., 2019a). The full pipeline of event-based sensors that output sparse data, stateful spiking neural networks which extract semantic meaning and asynchronous hardware backends allows for large gains in power-efficiency when compared to conventional systems.

## 1.2 ADVERSARIAL ATTACKS ON DISCRETE DATA

The history of attack strategies against various kinds of machine-learning algorithms pre-dates the advent of deep learning (Biggio & Roli, 2018), but the phenomenon received widespread interest when adversarial examples were first found for deep convolutional networks (Szegedy et al., 2013). Generally speaking, given a neural network classifier $C$ and an input $x$ which is correctly classified, finding an adversarial perturbation means finding the smallest $\delta$ such that $C(x + \delta) \neq C(x)$. Here, "smallest" refers to minimising $\|\delta\|$, where the norm is chosen arbitrarily depending on the requirements of the experiment. For example, using the $L^\infty$ norm (maximum norm) will generally make the perturbation less noticeable to a human eye, since the difference in any pixel value between the original and perturbed images will be below a maximum value that is kept as low as possible. Conversely, the use of the $L^1$ norm will encourage sparsity, i.e. a smaller number of perturbed pixels. The main challenges in transferring existing adversarial algorithms to event-based neuromorphic vision lie in the dynamics of the data and network, which develop in continuous time, and in the discrete nature of events, which can either be present or absent at a given time and location, unlike the continuous pixel values of traditional image data.

Event-based sensors encode information in the timing, location, and polarity of events, which can be of 'on' or 'off' type. Because at any point in time an event can either be triggered or not, one can simply view event-based inputs as binary data by discretising time (Figure 1). In this view, the network's input is a three-dimensional array whose entries describe the number of events at a location $(x, y)$ and in time bin $t$; an additional dimension, of length 2, is added due to the polarity of events. If the time discretisation is sufficiently precise, and no more than one event appears in each bin, the data can be treated as binary. A possible approach to attacking these data is exploiting recent work done on attacking binary images, i.e. with either black or white pixels, which are used in the automatic processing of cheques and other documents. Most methods proposed for attacking binary inputs have focused on brute-force approaches that rely on heuristics to reduce the search space (Bagheri et al., 2018; Balkanski et al., 2020). For example, SCAR (Balkanski et al., 2020) is a black-box algorithm that only assumes access to the output probabilities of the network. The algorithm flips bits in areas chosen according to a specific heuristic and keeps flipped those that cause a change in the confidence of the network. Naturally, this algorithm does not scale well to large input sizes, as the number of queries made to the network grows exponentially. In particular, this becomes a serious problem when the time dimension is added, greatly increasing the dimensionality of the

input. Instead, in this paper, we chose to focus on the easier problem of white box attacks, where the attacker has full access to the network and can backpropagate gradients through it. This allows us to adapt faster and more effective algorithms to the case of event-based data.

To this end, we chose to adapt existing attack strategies so that they could work with the time dynamics of spiking neural networks, and with the discrete nature of event-based data. We test our attacks on the Neuromorphic MNIST (Orchard et al., 2015) and IBM Gestures (Amir et al., 2017) datasets, which are the most common benchmark datasets within the neuromorphic community. Previous work on adversarial attacks in spiking networks has been reported by Sharmin et al. (2020); however, their work only uses static image data with continuous pixel values converted to Poisson input frequencies, so does not involve dealing with discrete data which was the main challenge in our work. More recently, Liang et al. (2020) did apply attacks to DVS data, using a discretised-gradient technique. They report high success rates, despite some notable problems of vanishing gradients. Concurrently with our work, Marchisio et al. (2021) designed custom algorithms for DVS data, rather than adapting existing ones, but did not report on the magnitudes of the resulting perturbations. None of these validated the effectiveness of their attack strategies against an on-chip model deployed on neuromorphic hardware. Our contributions beyond the existing literature can be summarised as follows:

- We provide detailed results to quantify the effectiveness and scalability of several adversarial attacks strategies, including some not tried before on SNNs.
- We show targeted universal attacks on event-based data in the form of adversarial patches, which do not require prior knowledge of the input.
- We validate the resulting adversarial examples on an SNN deployed on a convolutional neuromorphic chip. To the best of our knowledge, this is the first time the effectiveness of adversarial examples is demonstrated directly on neuromorphic hardware.

## 2 METHODS

### 2.1 ATTACK STRATEGIES

**Projected Gradient Descent**   As a baseline, we use Projected Gradient Descent (PGD) (Madry et al., 2019), a standard attack algorithm which we use on discrete data in two ways. The first consists in naively rounding the data at each iteration. However, in this case, updates will be retained only if the gradient magnitude is large enough: otherwise, the small changes made to the adversarial input are lost due to the subsequent discretization. Instead, we adopt an approach that prevents this loss of information: we keep a *continuous* version of the image as a copy, but use the gradients computed on the *discretized* image to update the continuous version which is kept in memory. To adapt PGD to the scenario where we want to find the smallest perturbation that triggers a misclassification, we sort the values based on how much PGD adjusted them. We then iterate through the sorted list of indices and flip each value until a misclassification is triggered. It should be noted that this step incurs most of the computational overhead, but is necessary to produce good results. Unless stated otherwise, we used the following values for the parameters: the magnitude of the initial random perturbation to the input is set to $\tau = 0.01$. The maximum norm of the perturbation was set to $\epsilon = 1.5$. We found that 50 iterations ($N_{\text{pgd}}$) of PGD sufficed and the results did not improve by much afterwards.

**Probabilistic PGD**   We also devised an alternative way of using PGD on discrete data, which we call "Probabilistic PGD". Probabilistic PGD works by assuming that the binary input was generated by sampling from a series of independent Bernoulli random variables. This approach aligns with how the DVS camera generates the binary data: the probability of emitting a spike at time $t$ is proportional to the light intensity, a continuous metric. For each round of PGD, the input is sampled in a differentiable manner by the Gumbel-softmax reparameterization trick (Jang et al., 2017):

$$\mathbf{x}_{\text{adv}} = \sigma\left(\left[\log(\mathbf{r}) - \log(\mathbf{1} - \mathbf{r}) + \log(\mathbf{p}_{\text{adv}}) - \log(\mathbf{1} - \mathbf{p}_{\text{adv}})\right]/T\right),$$

where $\mathbf{r} \sim \mathcal{U}(\mathbf{0}, \mathbf{1})$, and $T = 0.01$ is a temperature parameter. The underlying probabilities $\mathbf{p}_{\text{adv}}$, instead of the pixel values $\mathbf{x}_{\text{adv}}$, are updated using the gradient obtained from the loss function that is minimised by PGD. We saw that this generally improved the performance compared to the PGD version explained above. Gradients are averaged over $N_{\text{mc}} = 10$ samples of $\mathbf{r}$. It should be noted that the need for a gradient sampling procedure significantly increases the runtime.

**SparseFool on discrete data**   To operate on event-based data efficiently, the ideal adversarial algorithm requires two main properties: sparsity and scalability. *Scalability* is needed because of the increased dimensionality given by the additional time dimension. *Sparsity* ensures that the number of events added or removed is kept to a minimum. One approach that combines the above is SparseFool (Modas et al., 2018), which iteratively finds the closest point in $L^2$ on the linearised decision boundary of the network using the DeepFool algorithm (Moosavi-Dezfooli et al., 2015) as a subroutine, followed by a linear solver that enforces sparsity and boundary constraints on the perturbation. Because Spiking Neural Networks (SNNs) have discrete outputs (the number of spikes over time for each output neuron), it is easier to incur in vanishing gradients as the perturbation approaches the decision boundary. Therefore, we had to make changes to the algorithm to take this into account. Firstly, we found that clamping the perturbation at every iteration of DeepFool, so that it was no smaller than a value $\eta$, offered protection against vanishing gradients. $\eta$ was treated as a hyperparameter that should be kept as small as it can without incurring in vanishing gradients. Secondly, to account for the discreteness of event-based data, we rounded the output of SparseFool to the nearest integer at each iteration. Finally, SparseFool normally involves upper and lower bounds $l$ and $u$ on pixel values (normally set, for images, to $l = 0; u = 255$). We exploit these to enforce the binary constraint on the data ($l = 0; u = 1$), or, in the on-chip experiments, to fix a maximum firing rate in each time bin, which is the same as that of the original input ($l = 0; u = \max(\text{input})$).

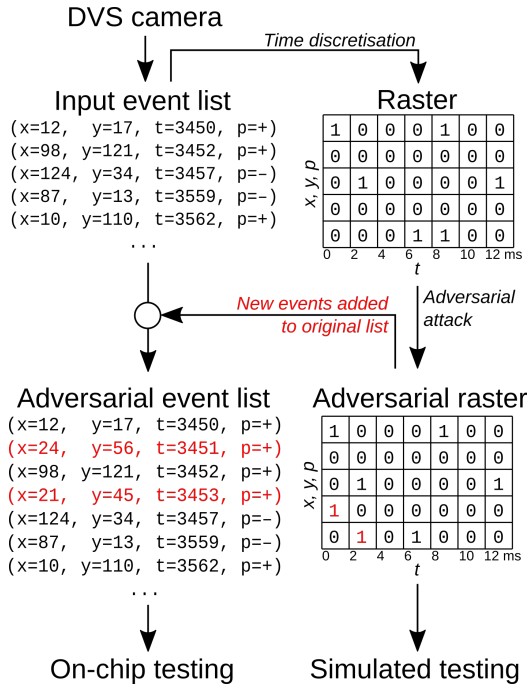

Figure 1: Schematic of the attack procedure on DVS data.

**Adversarial patches**   As the name suggests, adversarial patches are perturbations that are accumulated in a certain region of the image. The idea is that these patches are generated in a way that enables the adversary to place them anywhere in the image. This attack is targeted to a desired label, and universal, i.e. not specific to an input. To test a more realistic scenario where an adversary could potentially perform an attack without previous knowledge of the input, we apply these patches to the IBM hand gesture dataset. We note that the prediction of the CNN trained on this dataset is mostly determined by spatial location of the input. For example, the original input of "Right Hand Wave" is not recognised as such if it is shifted or rotated by a substantial amount. In order to simulate effective realistic attacks, we choose to limit both computed and random attack patches to the area of where the actual gesture is performed. As in Brown et al. (2017), we generate the patches using PGD on the log softmax value of the target output neuron. PGD is performed iteratively on different images of the training set and the position of the patch is randomised after each sample. For each item in the training data, the algorithm updates the patch until the target label confidence has reached a pre-defined threshold. The algorithm skips the point if the original label equals the target label. This process is repeated for every training sample and for multiple epochs. To measure the effectiveness of our computed patches, we also generate random patches of the same size, and measure the target success rates. In a random patch, every pixel has a 50% chance of emitting a spike at each time step.

## 2.2   DATASETS AND DATA PREPARATION

**Binarised MNIST**   We tried our methods on three datasets. The first is a binarised version of MNIST (BMNIST for short), which is derived from the popular MNIST Handwritten Digits database (LeCun & Cortes, 2010), binarised so that pixel values 0 to 127 are mapped to white, and 128 to 255 are mapped to black. No other preprocessing is applied. This is *not* a dataset of DVS recordings: we use it in order to compare our white box attacks against the SCAR attacks for binary datasets mentioned above (Balkanski et al., 2020).

**Neuromorphic MNIST**   Our first DVS benchmark is NMNIST (Neuromorphic MNIST), which consists of 300 ms-long recordings of MNIST digits that are captured using the saccadic motion of a DVS sensor (Orchard et al., 2015). This is the most commonly used DVS benchmark dataset for simpler tasks: since digits are only translating through the frame without changing, temporal features are not necessary for classification. When testing the spiking network, and for creating adversarial examples, each sample is fed to the network as a sequence of 5 ms-long binary frames. Additional spikes that fall in the same pixel within the same 5 ms window are discarded, so that each bin can contain either 0 or 1 events per pixel. The resulting data is a binary array (referred to as "raster") of dimensions $(t, p, x, y) = (60, 2, 34, 34)$, where $t = 300\,\text{ms}/5\,\text{ms} = 60$ is the number of time bins, $p = 2$ are the polarity channels, and $x = y = 34$ is the spatial resolution of the recording.

**IBM Gestures**   For a more advanced event-based vision benchmark, we used the IBM Gestures dataset, which consists of recordings of 11 classes of human gestures, captured under three different lighting conditions (Amir et al., 2017). Here, unlike the previous cases, the model must have some ability to process features in time, e.g. to distinguish between clockwise and counterclockwise hand motion in the same spatial position. The length of each gesture recording varies between 4 and 7 seconds. In this work, we never test on the full length of the recording at once, but we use 200 ms slices as the fundamental unit of the dataset. The data fed to the spiking network at test time are the same 200 ms samples, with time discretised in 10 ms bins. As above, spikes are capped to 1 per pixel per time bin. The dimensions of the resulting raster are $(t, p, x, y) = (20, 2, 128, 128)$. The experiments designed to run on the chip were binned at a higher time resolution of 2 ms since the neuromorphic hardware is capable to process events in continuous time.

## 2.3   NETWORKS

For the BMNIST experiments, we use a non-spiking network, similar to the one used in Balkanski et al. (2020): two $3 \times 3$ convolutional layers (32 and 64 channels each), with ReLU activations, followed by $2 \times 2$ max-pooling, dropout, and a fully connected layer of 128 features, projecting onto the final layer of 10 output units. The network is trained for 50 epochs at batch size 64, using the Adam (Kingma & Ba, 2014) optimiser with learning rate $10^{-3}$ on a cross-entropy loss function. The network reached a test accuracy of 99.12%.

The spiking networks used for the NMNIST and IBM Gestures tasks are simulated using a PyTorch-based SNN library which simulates non-leaky, linear integrate-and-fire neurons with no synaptic dynamics, equivalent to the ones emulated by the neuromorphic chip. In this neuron model, the inputs to each neuron are multiplied by the input weight and simply added to the neuron's membrane potential. The neuron spikes as soon as its membrane potential reaches a threshold, which is always set to 1. The threshold value is then subtracted from the membrane potential. The network's output label is the one corresponding to the output neuron that spikes the most, over the timespan during which the input is presented. There are no bias terms in our SNN's convolutional and linear layers.

The models used for NMNIST were trained using the "weight transfer" method, whereby an equivalent CNN is trained on accumulated frames (i.e. summing the data over the time dimension), and the CNN weights are transferred to the spiking network with thresholds set to 1 (Rueckauer et al., 2017; Sorbaro et al., 2020). The ANN was trained with Adam at batch size 64 with learning rate $10^{-3}$ for 10 epochs. We then rescaled the weights by layer-wise global factors so that the 99th percentile of activity was the same at each layer, as described by Rueckauer et al. (2017). The model we used consists of three convolutional layers of 20, 32, and 128 channels (kernel size 5 for the first, 3 for the other two), each followed by ReLU activation and $2 \times 2$ average-pooling. The convolutional stack is followed by a fully connected layer with feature size 500, which projects onto the 10 output units. The network achieves 84.93% classification test accuracy.

For the IBM Gestures task, training is done using backpropagation-through-time (BPTT), required even for feed-forward networks, because of the neurons' internal states, which persist in time. We make use of a surrogate gradient in the backwards pass to enable learning despite the discontinuous nature of spikes (Neftci et al., 2019): for gradient purposes, the neuron's nonlinearity is treated as a ReLU with zero-point placed at a value *threshold – window*. The window value is set to 0.5. For the simulated experiments, we used a network with a convolutional layer of kernel size 2, stride 2, and 8 channels, followed by two convolutional layers of kernel size 3 and 8 channels, and a fully connected layer of 64 channels that projects to the 11 output units. After every convolutional layer,

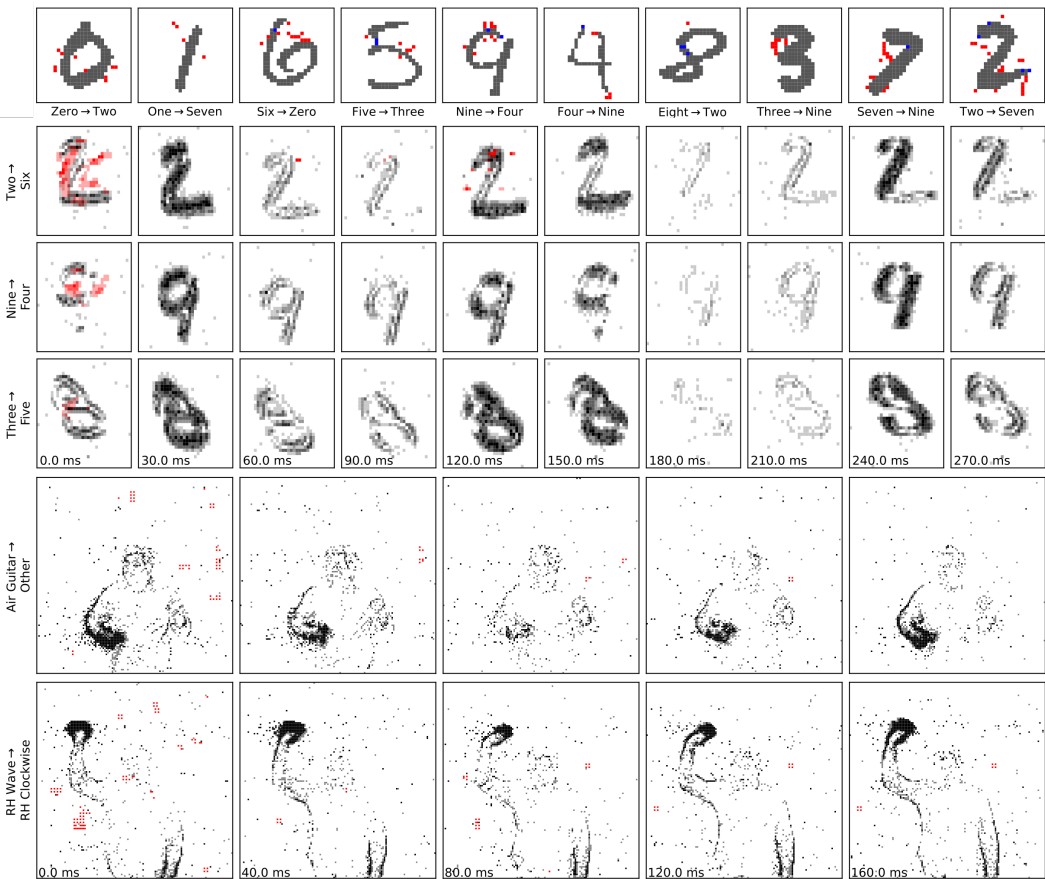

Figure 2: Examples of adversarial inputs on the BMNIST (top), NMNIST (middle) and IBM Gestures (bottom) datasets, as obtained by the SparseFool method. The captions show the original (true) label, correctly identified, and the class later identified by the model. The data was re-framed in time for convenience of visualisation. Red indicates added spikes. In the BMNIST examples, blue indicates removed pixels. We note that in the lower-dimensional BMNIST case, the effect of the attack is semantically interpretable: for example, adding a stroke that closes the upper left part of a "7" makes it look like a "9" not only for the network but also for a human observer. See the supplementary video for more examples and motion visualisation.

batch-norm, spiking activation, and $2 \times 2$ average pooling are inserted. This network achieves a classification test accuracy of 84.2%. The network used for the on-chip experiments has a slightly different architecture and does not have batch-normalisation layers to make it compliant with the hardware.

## 2.4 EXPERIMENTS ON THE NEUROMORPHIC CHIP

In order to verify our attack strategies in a more realistic scenario, we ran our experiments on neuromorphic hardware[1], which is especially suited for SNN inference due to its asynchronous nature. We use a digital, convolutional neuromorphic chip designed for computer vision applications. Weight precision, number of computations per second and throughput are typically reduced as the hardware is optimised for very low power consumption. This can lead to a degradation in prediction accuracy when compared to simulations. Because the networks detailed in the previous sections have to be modified in order to make them suitable for neuromorphic on-chip inference, their weights are rescaled and discretised as required by the chip's 8-bit weight precision.

---

[1]The name and brand of the chip in question have been redacted in this version for anonymisation purposes, and will be included in the final version of the paper.

| | Attack | Success Rate (%) | Median Elapsed Time (s/sample) | Median No. Queries | Median $L^0$ |
|---|---|---|---|---|---|
| **BMNIST** | SCAR | **100.00** | 1.14 | 1175 | **7** |
| | PGD | 98.89 | 0.16 | 102 | 50 |
| | Probabilistic PGD | 99.70 | 0.54 | 275 | 23 |
| | SparseFool ($\eta = 0.2, \lambda = 2$) | 99.90 | **0.08** | **11** | 14 |
| **NMNIST** | PGD | 48.63 | 72.56 | 1052 | $-^\dagger$ |
| | Probabilistic PGD | 54.46 | 68.35 | 774 | 522 |
| | SparseFool ($\eta = 0.2, \lambda = 2$) | 99.76 | 30.22 | 45 | **254** |
| | SparseFool ($\eta = 0.5, \lambda = 2$) | **99.88** | **13.08** | **26** | 268 |
| **IBM** | SparseFool ($\eta = 0.1, \lambda = 3$) | **100.00** | 2.78 | **11** | 310 |
| | SparseFool ($\eta = 0.1, \lambda = 2$) | 99.87 | **2.57** | **11** | 200 |
| | SparseFool ($\eta = 0.1, \lambda = 1$) | 97.69 | 3.02 | 17 | **116** |

$^\dagger$ Samples for which the attack was unsuccessful were considered to have $L^0 =$ undefined. Because PGD fails more than half of the time, the median is undefined.

Table 1: Comparison of attack strategies (1000 samples). SCAR was implemented according to the pseudo code in Balkanski et al. (2020) and PGD was run for 50 iterations. SparseFool takes only a fraction of the time compared to PGD while obtaining much sparser results at almost perfect success rate on Neuromorphic MNIST. The input size for this dataset is set to (60,2,34,34). We also use SparseFool to attack samples from the IBM Gestures dataset at different values of $\lambda$, a parameter trading-off speed and sparsity. The success rate is here defined as the fraction of samples that were initially correctly classified, for which the attack algorithm converged to an adversarial example that the network classifies incorrectly.

As this work focuses on white-box attacks, we first computed the adversarial examples using the network simulation on the computer, then tested both original and attacked spiketrains in simulation and on the chip. The simulation and the attack work in discrete time, while the chip receives events in continuous time. In order to convert the discrete-time attacked raster back to a list of events for the chip, we compared the original and attacked rasters, identifying new events added by the attack and adding them to the original list (Figure 1). We empirically found very few events removed by SparseFool (see supplementary section 1) and chose to ignore removals for on-chip experiments.

## 3 RESULTS

### 3.1 SPARSEFOOL ATTACKS ON BINARY AND DVS DATA

Table 1 compares the different algorithms on Binary-MNIST and shows that SparseFool finds successful adversarial examples with a low median $L^0$ (i.e. number of perturbed pixels), while requiring a very low median execution time. Figure 2 (top) illustrates samples of perturbations found by SparseFool and the corresponding label that was predicted by the network after applying the perturbation. Because of the small sample size and the fact that there is no time dimension, Binary-MNIST enables us to compare SparseFool to other, more inefficient methods. However, more realistic datasets are needed to truly evaluate the feasibility of applying these algorithms.

After having established that SparseFool can efficiently and reliably generate sparse perturbations on discrete data, we evaluated SparseFool's performance on the DVS benchmarks for different hyperparameters (Table 1). $\eta$ indicates the minimum step size for updates to the perturbation: higher values of $\eta$ find less precise perturbations (larger $L^0$ values), but are sometimes needed in order to prevent zero-gradient issues within the algorithm. $\lambda$ is the sparsity parameter: lower $\lambda$ (with a minimum of 1) yields sparser results, but gives a slightly lower success rate. Overall we consistently found that SparseFool performs better than PGD and Probabilistic PGD, both in terms of success rate and in the number of added or suppressed events. Additionally, it requires far fewer iterations and therefore converges more quickly. Figure 2 and the supplementary video show examples of successful attacks. Supplementary section 1 provides further information on the characteristics of the resulting samples.

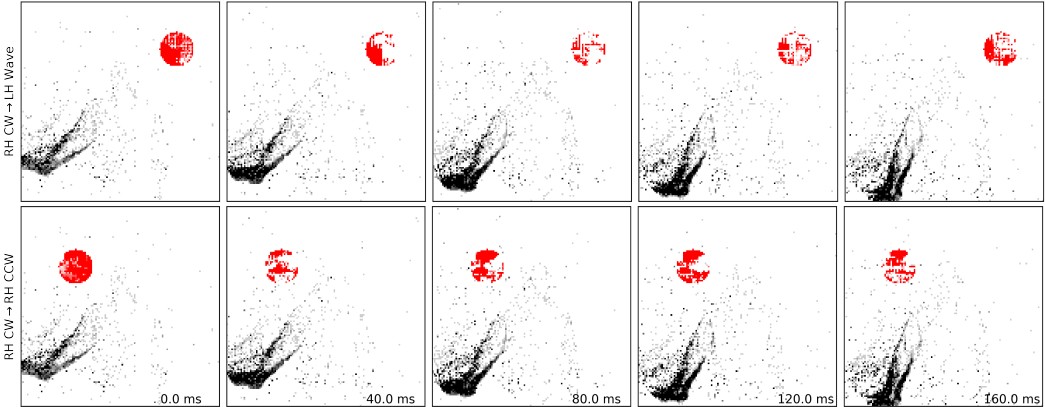

Figure 3: Examples of adversarial patches successfully applied to a single "right hand clockwise" data sample, with different target classes. See also the supplementary video for motion visualisation and more examples of successful patch attacks.

## 3.2 VALIDATION ON NEUROMORPHIC HARDWARE

We randomly chose 1000 snippets, each 200 ms long, from the IBM Gestures dataset, on which we ran the SparseFool attack. Out of these, 833 were successfully classified by the network and were therefore eligible for an attack; the attacks converged and were successful in simulation in 777 cases. We presented these 777 successful attacks to the chip, alongside the un-attacked original data, finding that 96.9% (753) of the originals are successfully classified by the chip, and 85.3% of the attacks are able to fool the chip (663) too. A possible reason for this discrepancy lies in how the chip is limited in computing capacity by weight quantization and restricted throughput per time unit, which causes some of the input events to be dropped. The conversion of binned data back into lists of spikes, discussed in the Methods section, is necessarily lossy at this time. In terms of attack efficiency, we observe a median $L^1$ distance (i.e., difference in number of spikes) of 903 among the attacks that were successful on chip, corresponding to a median 9.3% increase in the number of spikes per sample. The full distribution is shown in Figure S1 (bottom left). Figure S1 also shows the time profile of the perturbation and how the network classified the data after the attack.

## 3.3 ADVERSARIAL PATCHES

Although we have demonstrated that one can achieve high success rates on custom spiking hardware that operates with microsecond precision, the applicability of this method is still limited, as the adversary needs to suppress and add events at high spatial and temporal resolution, thus making the assumption that the adversary can modify the event-stream coming from the DVS camera. Furthermore, SparseFool assumes knowledge of the model and requires computing the perturbation offline, which is not feasible in a timely manner. In a more realistic setting, the adversary is assumed to generate perturbations by changing the input the DVS camera receives on the fly.

Using the training data from the IBM Gestures dataset, we generated an adversarial patch for each target class with high temporal precision (event samples of 200 ms were binned using 0.5 ms-wide bins) and evaluated the effectiveness in triggering a targeted misclassification both in simulation and on-chip using the test data. To simulate spatial imprecision during deployment, each test sample was perturbed by a patch that was randomly placed within the area of the original gesture. Table 2 summarises our findings on target success rates for generated and random patches. Simulated results show high success rates, and on-chip performance shows a slight degradation, which can be expected due to weight quantization on the tested specialised hardware. We also found that the chip had trouble processing inputs because most of the added patch events occurred concentrated in the beginning of recordings in a large transient peak. In one case, the targeted attack for label "Arm Roll" mostly fails on chip as not all events are processed, which makes it harder to discriminate between similar labels such as "Hand Clap", a similar gesture that occurs in the same central spatial

| Target label | Hand clap | RH Wave | LH Wave | RH Clockwise | RH Counter Clockwise | LH Clockwise | LH Counter Clockwise | Arm Roll | Air Drum | Air Guitar | Other |
|---|---|---|---|---|---|---|---|---|---|---|---|
| Adversarial patch | 90.3 | 99.0 | 89.8 | 87.3 | 79.7 | 49.7 | 51.5 | 63.6 | 79.1 | 92.3 | 64.7 |
| Adv. patch (on-chip) | 94.0 | 89.0 | 94.1 | 81.3 | 65.1 | 35.9 | 43.8 | 5.0 | 82.7 | 87.3 | 66.8 |
| Random patch | 18.8 | 80.7 | 77.0 | 0 | 0 | 3.6 | 0.6 | 0 | 0 | 12.6 | 16.6 |
| Rand. patch (on-chip) | 43 | 76.8 | 72.2 | 0 | 0 | 9.0 | 2.4 | 0 | 0 | 0 | 17.7 |

Table 2: Adversarial patches for different target labels were evaluated on– and off–chip. Shown here are the success rates in percent for each target label. An attack is considered successful if the original label is not the target label and the network predicts the target label when the patch is applied.

location. This could somewhat be mitigated by limiting the number of events in a patch to ensure that they could all be correctly processed on the chip.

We compare this result with a baseline of randomly generated patches, and we observe that two labels, namely "Left" and "Right Hand Wave" subsume all other attacked labels in this case. This hints that randomly injecting events in various locations is not enough to perturb network prediction to a desired label and that our algorithm succeeds in finding a meaningful patch. To summarise, adversarial patches are effective in triggering a targeted misclassification both on– and off–chip compared to randomly generated ones. Figure 3 and the supplementary video show examples of successful patch attacks. Importantly, these attacks are universal, meaning that they can be applied to any input and do not need to be generated for each sample.

## 4 DISCUSSION

We studied the possibility of fooling spiking neural networks through adversarial perturbations to dynamic vision sensor data, and verified these perturbations on a convolutional neuromorphic chip. There were two main challenges to this endeavour: the discrete nature of event-based data, and their dependence on time. This translated, in practice, in the need for an extra temporal dimension, and in different sparsity requirements, because the magnitude of the perturbation is measured in terms of number of events added or removed. For this purpose, we adapted the sparse adversarial algorithm SparseFool, and showed that it achieves high convergence rates on time-discretised samples of the Neuromorphic MNIST and IBM Gestures datasets. Empirically, we observe that the algorithm mostly resorts to adding, rather than removing, input events, and the number of new events necessary to fool the network varies significantly from sample to sample. In the best cases, the attack requires the addition of less than a hundred events over a 200 ms sample, an increase of a few percent. With this, we have proven adversarial examples in DVS data are possible, and, to the best of our knowledge, we were also the first to show that the perturbation is effective in a network deployed on a neuromorphic chip. As the history of adversarial attack algorithms shows, future research in this field may well find adversarial perturbations that are even less noticeable, detectable, or computationally intensive. One should be aware of this possibility when deploying any neuromorphic devices using DVS technology together with neural network models in all contexts where malicious attacks may have serious consequences, such as autonomous driving, surveillance, or control. For this reason, it is also important to consider how to counter these attacks. Defence mechanisms such as adversarial-aware training exist and can provide better robustness. In preliminary work outlined in supplementary section 2 we apply one such method to SparseFool attacks.

SparseFool computes perturbations offline, and it is currently not obvious how to do this on the fly on a live stream of DVS events. Therefore, we also investigated a more realistic setting, where an adversary can, with low spatial, but high temporal precision, inject spurious events in the form of a patch inserted into the visual field of the DVS camera. We showed that we can generate patches for different target labels, which trigger targeted misclassifications with high precision. Although these patches require a much higher amount of added events, they do not require prior knowledge of the input sample and therefore offer a realistic way of fooling deployed convolutional neuromorphic systems. A natural next step would be to understand whether it is possible to build real-world patches that can fool networks when shown to the DVS camera from a variety of distances and orientations, as Eykholt et al. (2018) did for photographs. Additionally, it will be interesting to see how important knowledge about the architecture is and if one can generate adversarial patches by having access to a network that differs from the deployed ones.

## 5 ETHICS STATEMENT

The authors aim to minimise the impact of potential attacks on deployed SNNs by contributing to overall understanding and supporting public discussion thereof.

A subset of the authors are currently employed by the neuromorphic chip manufacturer, which could be perceived as a conflict of interest. We did our best to counteract this by making available all source code used in the experiments and we encourage other researchers to reproduce our results. The other authors report no current conflicts of interest.

## 6 REPRODUCIBILITY STATEMENT

The authors made available all code that was used to generate results and plots in this paper, which we attach as supplementary material. The code will be made publicly available after the review process. The libraries used for SNN simulation and DVS data management are available as open-source code. The DVS datasets are available online. The neuromorphic chip is available for research purposes. When reproducing experiments on chip, results are expected to differ slightly due to variations in the manufacturing process.

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

SUPPLEMENTARY MATERIAL

# 1 EMPIRICAL ANALYSIS OF THE RESULTING SPARSEFOOL PERTURBATIONS

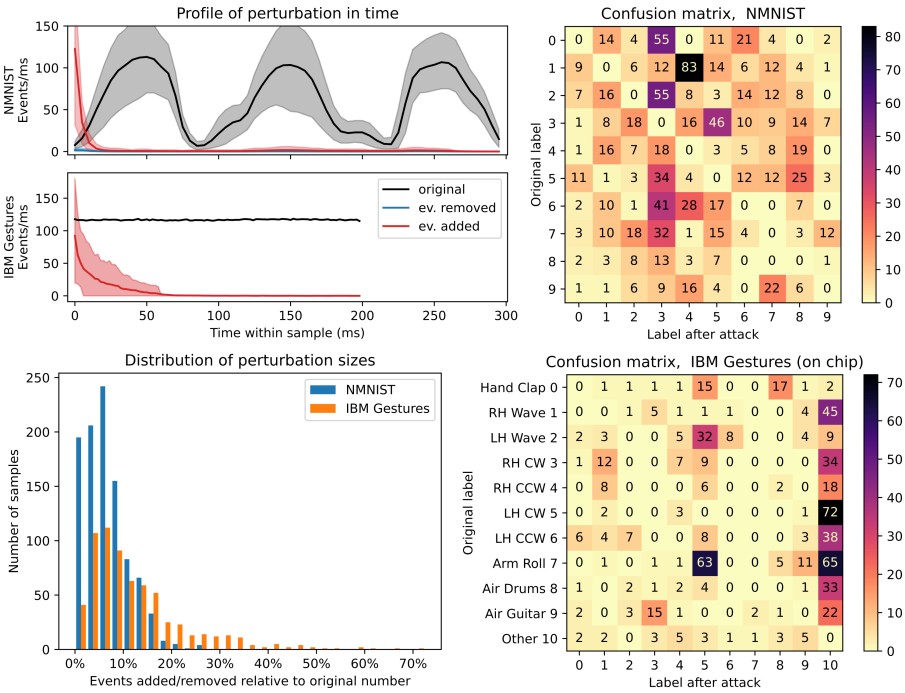

Figure S1: Properties of the adversarial perturbations found by SparseFool, for two experiments: NMNIST (in simulation, $\eta = 0.5$, $\lambda = 2$) and IBM Gestures (as tested on chip). *Top left*: Number of events in time within each data sample. The shaded areas represent the 0.1-0.9 interquantile range (not shown for the 'original' curve in the bottom panel). The perturbation tends to consist of spikes added at the beginning of the sample, especially for NMNIST which does not rely on temporal structure for inference. Very few spikes are removed, which justifies the choice of ignoring removed spikes in on-chip experiments. The periodic structure of NMNIST samples is intrinsic to the dataset, recorded with saccades. *Bottom left*: Distribution of increase in number of spikes after the attack, relative to the original number. *Right*: Matrices showing the label identified by the network when presented with the adversarial examples, given the original label, for the two experiments. Most IBM Gestures classes are perturbed towards the 'other' class, while there is no clear structure in the NMNIST case. LH = Left Hand, RH = Right Hand, (C)CW = (Counter) ClockWise.

With the aim of gaining more insight into the behaviour of our methods, we studied the characteristics of the perturbations resulting from SparseFool attacks in more detail. For this, we chose two specific experiments: a SparseFool run on NMNIST with hyperparameters $\eta = 0.5$ and $\lambda = 2$; and the on-chip IBM Gestures experiment. First, we empirically notice that SparseFool-based perturbations rarely involve the removal of events. In the NMNIST experiment considered here, an average of 7.6 events is removed from each sample, compared to an average of 214 spikes added. This justified our choice to ignore removed events in the course of the on-chip experiments.

As is evident from the examples in Figure 2, we also find that SparseFool's adversarial perturbations tend to consist in the insertion of spikes at the beginning of the sample, with only a few spikes added later in time. The top left panel of figure S1 shows the time profile of the perturbations in detail. We believe this is a consequence of the use of the non-leaky neuron model. In non-leaky neurons, information can be stored indefinitely in the membrane potential, so early spikes have a further chance of contributing to a spike later in time, and are more effective compared to events added later in the sample. This effect is also present in the IBM Gestures experiment, but looks less prominent, possibly because networks trained with BPTT on data with richer features in time have a non-trivial dynamics. In this sense, we expect this phenomenon to be further reduced or

disappear entirely when the task is strictly linked to the time evolution of the input signal, such as in auditory speech recognition. The timing of adversarial events could potentially be used for model interpretability purposes, to measure how much the model relies on temporal features.

Further to the median values reported in table 1, the lower left panel of figure S1 reports the full distributions of the number of added or removed events ($L^1$ distances). Here, we display the numbers relative to the original number of events in the sample. We notice a minority of cases where the attack is successful only at the cost of a very significant injection of events.

Finally, we analysed the statistics of classes identified by the networks after the attack. SparseFool is used as an "untargeted" algorithm, i.e. it attempts to change the output of the network but without requirements on what the new class should be. Unsurprisingly, the "other gesture" class is a natural target class for many ground truth classes, but there are some exceptions which we find rather natural, such as "left hand wave" gestures being most often converted to "left hand clockwise". Conversely, we observe no dominant target class in the NMNIST experiment. If the target class structure is undesirable, targeted attacks can be used instead.

## 2   DEFENCE VIA ADVERSARIAL TRAINING

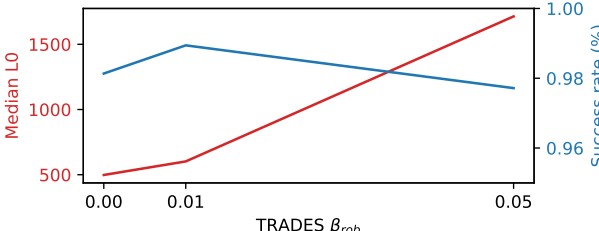

Figure S2: Success rate and median $L^0$ of SparseFool for networks trained with TRADES robustness based on PGD attacks.

Once it is known that a model or system is sensitive to a certain type of adversarial attack, it is natural to investigate whether there is a way to build a network that is more resistent to these attacks. We therefore experimented with adversarial training using the TRadeoff-inspired Adversarial DEfense via Surrogate-loss minimization (TRADES) method (Zhang et al., 2019). The method consists of adding a new term to the loss function during training, which minimises the Kullback-Leibler divergence between the output of the network when the original input is presented, and the output when the adversarial example is presented:

$$\mathcal{L}_{\text{rob}} = \mathcal{L} + \frac{\beta_{\text{rob}}}{B} \, \mathrm{D}_{\mathrm{KL}}(f(\mathbf{x}_{\text{adv}}); f(\mathbf{x}_0)).$$

Here, $B$ is the batch size, $\beta_{\text{rob}}$ is the parameter that defines the trade-off between robustness and accuracy, $f$ is the network and $\mathbf{x}_{\text{adv}}$ is the adversarial input. Although networks that were trained using SparseFool would probably be more robust, we opted for PGD at training time, since it can be easily batched — but we attack the resulting networks using SparseFool. We used PGD in the $L^\infty$ domain and chose $\epsilon = 0.5$ as the maximum perturbation, with $N_{\text{pgd}} = 5$ attack steps. We also did not greedily chose the best indices to flip as described in section 2.1. Even if this was a much simplified version of the PGD attack, we found that this configuration produced perturbations with reasonable Hamming distances while being extremely efficient, and it was sufficient in inducing some level of robustness.

From the results in Figure S2 we note that the success rate is still quite high despite the adversarial training for the choices of $\beta_{\text{rob}}$ we considered. However, given the fact that SparseFool aims at finding the smallest perturbation that triggers a misclassification, this is expected, and there is already a noticeable increase in the number of added spikes required, which is indeed a sign of robustness. In other words, the adversarially-trained network requires stronger and less stealthy attacks before it is fooled. Further work is required for a comprehensive investigation of other possible defence strategies.

