# OpenReview forum: "Adversarial Attacks on Spiking Convolutional Networks for Event-based Vision"
_ICLR.cc/2022/Conference — ICLR 2022 Submitted_

### Official Review · Reviewer_YwAR · 2021-10-29

**Correctness:** 3
**Technical Novelty And Significance:** 2
**Empirical Novelty And Significance:** 3
**Recommendation:** 5
**Confidence:** 5

**Main Review:**

The paper is nicely written, and presented experiments and visualizations are well organized. Empirical validations of the proposed methods on neuromorphic hardware add a significant value to the paper with respect to the state-of-the-art.

On the other hand, methodological contribution of the paper is rather shallow, and Section 2.1 is not describing the used methods in detail. It should be clearer in mathematically describing the attack optimization objective within the SNN setting, e.g., on page 4 while describing SparseFool, some variables are mentioned out of context. Most of the important content on what is actually done for the attacks is described shortly in text and in the Supplementary algorithms. Overall, the authors implemented SparseFool attacks of [Modas et al, 2018] for SNNs, and demonstrated that SNNs with event-based inputs can be adversarially attacked. However authors also do not evaluate their attacks in comparison to the state-of-the-art PGD-like attacks with spike-compatible gradient proposed for SNNs by [Liang et al, 2020]. In fact this study is also not correctly referenced in the paper, mentioned as a work that utilizes: “… brute-force approaches that rely on heuristics to reduce the search space [Liang et al, 2020, …]”, which is not the case.

I think evaluating a binary image recognition CNN on the BMNIST dataset with SCAR attacks is not a good choice of a baseline for this work, and seems distant from the main purpose of the paper. It would be more suitable to compare the same SNNs and proposed attacks with previously presented PGD-variant attacks for event-based data [Liang et al., 2020].

One of the earliest studies in this domain by Sharmin et al, "Inherent adversarial robustness of deep spiking neural networks: Effects of discrete input encoding and non-linear activations.”, ECCV 2020, is also missing and should be discussed in the paper. In some ways, the PGD attacks presented by [Sharmin et al., 2020] for SNNs resembles to the standard PGD approach demonstrated in the paper (Section 1.2 of the Supplementary Materials). This could be highlighted and discussed in detail with the revisions. For instance, N-MNIST experiments are performed with models trained using ANN to SNN weight transfer, which [Sharmin et al., 2020] similarly demonstrated with their PGD-like attacks.

Results presented in Table 2 (and the supplementary video) are very interesting. It seems like a random patch perturbation can be quite effective for a RH Wave label, but not at all for RH Clockwise or RH Counter Clockwise labels. Could this be a result of the relevant temporal information constructing a significant aspect of the attacking patches? At this point, what I think is another missing “random patch baseline” to compare against, is one that can exploit temporal information. For example, if one would invert the signs of the events of learned adversarial patches before addition (i.e., using un-meaningful inverses of the learned adversarial patches), would it already be a better attack than a naive random patch (which only exploits locations but not temporal information)?

Looking at Fig 2 IBM Gestures examples, it looks like perturbation events that are much outside the region of movement are even sufficient to lead to misclassification (i.e., models are very vulnerable to arbitrary events occurring outside the relevant region). To compare this vulnerability against some potential robustness gains, I would be curious if the authors implemented any simple robust training approach that resembles to adversarial training? i.e., would the models be still very robust to additive events occurring at irrelevant locations? These observations can be discussed in the revisions.

Minor comments:
- Authors report that the SNN used for IBM Gestures has a test accuracy of 84.2%. For completeness, can the authors also report what is the on-chip test accuracy of the model following the modifications (e.g., removing batch-norm) and weight precision quantizations to make the model compatible with the hardware?
- Please write in the caption of Fig 2 that those examples of adversarial inputs are crafted via SparseFool attacks (the reader discovers it much later in the text).
- Caption of Algorithm 2 in the Supplementary Materials includes some interesting information that could be extended. Authors state that 50 iterations of PGD was sufficient and results did not improve afterwards. It would be interesting to plot this observation in the S.M. as it seems to demonstrate some inherent robustness.

**Summary Of The Paper:**

Authors present white-box adversarial attack algorithms for spiking neural networks (SNNs) with event-based visual data inputs, and experimentally demonstrate vulnerability of SNNs in various settings. Extensions of existing adversarial attacks on convolutional networks (i.e., PGD and SparseFool) are presented and utilized in a setting for SNNs with event-based inputs. Experimental evaluations are performed on the Neuromorphic-MNIST and IBM Gestures dynamic vision sensor datasets, and validations of these attacks on neuromorphic hardware are performed.


**Summary Of The Review:**

The paper is nicely written and organized from an empirical contribution perspective. However the paper is rather weak in depicting its methodological contributions in my opinion, and lacks comparisons/references to relevant SOTA works. I listed my major concerns in the main review, and would be willing to re-address my rating based on the authors’ responses and revisions.

---

> ### Author Response · Authors · 2021-11-22
> **Addressing comments by reviewer YwAR**
>
> Again we would like to thank the reviewer for their respectful, clear, and fair review.
>
> Regarding the clarity of the methods sections, and in particular of how the algorithms are applied to DVS data, this was pointed out also by other reviewers. We have therefore rephrased and expanded the algorithms descriptions in section 2.1, and moved more details from the supplementary material to this section. We also removed the algorithm schematics from the supplementary material when they were redundant with the algorithm description in the existing literature.
>
> We thank the reviewer for bringing the Sharmin et al. 2020 paper to our attention, as we were not aware of it. The difference they report between the adversarial robustness of spiking networks trained by ANN to SNN weight transfer and by backpropagation through time is extremely interesting. However, there is a very significant difference between our work and theirs: the biggest challenge in our work was dealing with DVS data, which are discrete. The Sharmin paper, however, deals only with traditional image data, which are fed to the network as Poisson spike trains of frequencies given by pixel values. This means they did not require any special techniques for updating the input data, since they could directly operate on the continuous pixel values, as is normally done in traditional computer vision. For this reason, we believe our results cannot be directly compared. Regardless, we do agree that this is a relevant paper and we have now cited it in the introduction.
>
> We apologise for the incorrect reference to the Liang paper, which is indeed unrelated to the heuristic and brute-force binary attacks. We had not considered this paper carefully enough. It’s in fact a very relevant contribution that already provides interesting results on adapting traditional attacks to DVS data. Compared to this work, 1. we use a different strategy for working on discrete data, which seems to work better against the vanishing gradient problem; 2. we use SparseFool, which is specifically designed to change the smallest number of spikes, unlike PGD and the like; 3. we test in the more challenging environment of a network deployed on the chip, which required a mechanism for re-translating the raster into a list of spikes; 4. we also try the patch attacks, which don’t require prior knowledge of the input and can be applied online. We updated the introduction to refer to this paper properly, and have rephrased the statement of our contributions.
>
> Regarding the use of temporal information in the patches attack, we looked at some preliminary results and note that flipping polarity in the generated patches results in severely degraded results, with the attack success rate close to zero for some of the target classes. It is possible, as the reviewer suggests, that this means the network exploits temporal information; however, temporal information exists in the data regardless of event polarity, and the networks trained with BPTT may be able to exploit this.
>
> Finally, we investigated whether it was possible to increase the robustness of the spiking CNNs with adversarial training. To do this, we used the widely-used TRADES algorithm (Zheng et al. 2019). TRADES adds a term to the natural loss that incurs a high loss when the network is susceptible to attacks in the input space. The two losses are balanced with a term that we denote $\beta_{rob}$ (see supplementary material), which we varied during our additional experiments. We found that by increasing this value (i.e. putting more emphasis on robustness), the networks that were trained were harder to attack by SparseFool: Networks trained with higher $\beta_{rob}$ required more than three times as many added events compared to the baseline. We have added these results in the supplementary material.
>
> About the  minor comments:
> * The accuracy is reported for the actual sample used for the tests: 833 out of 1000 where classified correctly (section 3.2).
> * Done, thanks for pointing this out.
> * This was an observation that we made when hand-tuning the value given the other hyper parameter values. We believe that this depends more on the choice of the other hyper parameters and is not a sign of inherent robustness.

---

> > ### Comment · Reviewer_YwAR · 2021-11-24
> > **Thanks for the responses**
> >
> > Thanks to the authors for their responses. I carefully read the revisions and comments.
> >
> > I can agree on the authors point of view on the differences with respect to the PGD attacks by Sharmin et al. However as the authors acknowledged, [Liang et al, 2020] is a suitable attack algorithm on DVS input data that is highly related with the current work. That being said, I can see the differences in the authors' methodology to explore novel attacks based on existing ones from ANN literature. However, given that the authors present their work from a perspective of "showing the effectiveness and scalability of several adversarial attacks", I believe then this novelty should be assessed with respect to the similar state-of-the-art machine learning methods for publication at ICLR.
> >
> > A minor comment on vanishing gradients with the attacks by [Liang et al, 2020]: I believe that paper actually proposes several mechanisms to particularly address this problem while crafting attacks for SNNs. Eventually this should be a strong baseline to compare the authors' contributions with adapted SparseFool attacks and adversarial patches.
> >
> > Minor comment on TRADES results: In my opinion the new results presented in Section S.2 of the Supplementary are equally important as the attack results presented in the main manuscript. It can be seen that PGD-based adversarial training of SNNs generalizes robustness to SparseFool attacks, while it is not clear to me which dataset and networks these evaluations belong to. As the authors also stated, these analyses certainly needs more depth in future work.
> >
> > Overall I decide to keep my score as it is, slightly below acceptance threshold.

---

### Official Review · Reviewer_FiWC · 2021-10-30

**Correctness:** 3
**Technical Novelty And Significance:** 3
**Empirical Novelty And Significance:** 3
**Recommendation:** 5
**Confidence:** 2

**Main Review:**

Strengths:
This paper draws a conclusion through rigorous and sufficient experiments, which provide quantitative results to prove the effectiveness of the attack algorithms on dvs data. How to deal with the discrete nature of event-based data and their dependence on time is explained clearly. Also, the authors verified the effectiveness of adversarial examples on neuromorphic chip for the first time.
Weaknesses:
1)More details about the existing attack algorithms after adapted and how the algorithms work on the discrete-time raster could be explained.
2)Some explanations about the meaning of the variable symbols such as ς ,r_i ,α_i and so on in the appended algorithms could be added.
3)More attack algorithms such as PGD and probabilistic PGD could be used on neuromorphic chip and calculate success rate in order to compare with the algorithm Sparsefool.


**Summary Of The Paper:**

This paper investigated how to adapt existing attack strategies in order to work with the discrete nature of event-based data. Also, the authors proved that Sparsefool can efficiently and reliably generate sparse perturbations on discrete data. Besides,the authors validate the adversarial examples on neuromorphic chip and the perturbation is effective after modified the networks. In a more realistic setting, the authors use adversarial patches to trigger a targeted misclassification. MNIST and IBM Gestures datasets are used to test the attack algorithms as a sequence of binary frames.

**Summary Of The Review:**

This paper verified the attack algorithms can be adapted to the dvs data. The Sparsefool algorithm achieves high success rates through adding and removing a few events. The authors proposed a next step, which building real-world patches that can fool networks.

---

> ### Author Response · Authors · 2021-11-22
> **Addressing comments by reviewer FiWC**
>
> We would like to generally thank all reviewers for their respectful, clear, and fair reviews. These reviews agree on several common points. We share many of these concerns, and we have tried to address them to the best of our abilities in these two weeks.
>
> Points 1) and 2) of this review, which relate to the lack of clarity in our explanations of how we adapt and run the algorithms to the DVS data, have been raised by other reviewers as well. We therefore decided to move more details about the algorithms to the main text, and have removed the algorithm schematics, which were out of context and with poor descriptions of mathematical symbols. For detailed mathematical descriptions we refer to the original papers these were taken from, and, instead, we explain the changes we made in text, in the main paper. Section 2.1 has therefore been significantly expanded.
>
> Regarding point 3), we did not run the PGD attacks on the chip because the chip with its inherent limitations will not perform better than results obtained in simulation. As such we chose the most successful attack strategy, which according to our simulations is generated by SparseFool.

---

### Official Review · Reviewer_jrcY · 2021-10-30

**Correctness:** 4
**Technical Novelty And Significance:** 2
**Empirical Novelty And Significance:** 2
**Recommendation:** 5
**Confidence:** 4

**Main Review:**

Strengths:
+ overall this is a solid paper, without unjustified claims, and with a straightforward message: adversarial attacks on event-based vision works
+ the paper is well written, the figures are clearly showing the effects of the adversarial attacks
+ a variety of different attacks is tried out, including adversarial patches
+ the paper is to the best of my knowledge the first to try out the effect of adversarial attacks on neuromorphic hardware

Weaknesses:
- as the authors note, there is concurrent work by Marchisio et al. 2021, which investigates a very similar question, and even goes beyond that by investigating potential defenses against adversarial attacks. The present paper only shows the possibility of attacks, but does not talk about defenses. Therefore the novelty is severely limited.
- the description of the adversarial attacks and their adaptation to the event-based domain is very short and could only be understood with the help of the supplementary material. I would propose expanding this description to make the main paper more self-contained.
- the results of the paper are not really surprising, and I would have hoped for more specific comparisons how adversarial attacks differ between conventional and event-based vision, and also some outlook into defenses

**Summary Of The Paper:**

The paper presents proof of the viability of adversarial attacks on CNN processing event-based vision data. The paper investigates adaptations of well-known white box attacks to the event-based and spiking domain, validates the claims on three public benchmarks, and investigates the effect of the adversarial attacks directly on neuromorphic hardware. The main result is that, just like for conventional computer vision, adversarial attacks can have a big effect on the prediction of CNNs for event-based vision inputs. The attacks also translate to networks run on neuromorphic hardware.

**Summary Of The Review:**

While the paper is formally correct and well written, there are other published papers that have shown the main claims of the paper, and have even gone some steps further by analyzing possible defenses against adversarial attacks. Therefore the novelty is too limited, and the key findings are not surprising. There are some new elements, e.g. the analysis of attacks on real neuromorphic hardware, but overall this is too little for acceptance at ICLR.

---

> ### Author Response · Authors · 2021-11-22
> **Addressing comments by reviewer jrcY**
>
> We would like to thank reviewer jrcY for their comments. We have addressed them as follows:
>
> This reviewer’s concern mostly seems related to the novelty of the paper, especially due to the lack of a discussion of defenses against the attacks we present. Despite the small amount of time available, we worked on implementing defence mechanisms during the review period and found that by using the TRADES algorithm, we could significantly boost robustness of our spiking CNNs. Specifically, we found that when our networks are trained with an additional robustness term that punishes susceptibility to PGD-based input attacks, SparseFool finds perturbations that are significantly larger compared to the baseline. However, the success rate of SparseFool remained high. For reasons of space, we added these results to the supplementary material.
>
> Concerning the work of Marchisio et al., beyond the defense mechanisms, we believe there are significant differences with our work:
> * Marchisio’s work does not build on previously existing adversarial algorithms, but creates custom ones for DVS data
> * For the majority of their attack strategies, they do not report figures regarding the number of spikes added or subtracted. This makes it difficult to quantitatively evaluate how visible and detectable the attacks are.
> * Their attack algorithms require prior knowledge of the input and are not universal like the patch attack, which makes them less relevant to a realistic use in a deployed context.
> * They do not test their attacks against a network deployed on neuromorphic hardware. This is an additional challenge because of the high temporal resolution at which chips work compared to simulations (requiring a conversion from rasters to lists of events), and, conversely, of the lower resolution in network parameters (weights are stored at low precision).
>
> Regarding the clarity of how we adapt the attack algorithms to the DVS data, this was indeed pointed out also by other reviewers. We have updated the paper so that the algorithms are explained in the main text rather than in the supplementary material, and the changes we made are more clearly expressed in text rather than in algorithm schematics.

---

> > ### Comment · Reviewer_jrcY · 2021-11-30
> > **Response to Authors**
> >
> > I´d like to thank the authors for their explanations and comments. I especially appreciate the efforts to include a discussion on defences against adversarial attacks, I think this is a promising direction and improves the paper. I will therefore raise my score from 3 to 5, but overall, given also the concerns of the other reviewers, I think this paper is still below the acceptance threshold.

---

### Official Review · Reviewer_2Mf5 · 2021-11-01

**Correctness:** 3
**Technical Novelty And Significance:** 2
**Empirical Novelty And Significance:** 2
**Recommendation:** 5
**Confidence:** 4

**Main Review:**

1.	It is not clear how the adversarial attack algorithms that are commonly used for discrete data are applied to event-based data in this work. Which are the design choices that are done to develop/modify these methods? Since these modifications constitute the main contributions of this work, they should be comprehensively discussed in the main manuscript, rather than in the supplementary material. The description in Sections 2.1 and 2.2 does not contain sufficient details.
2.	In this work, the novelty is low because the authors use already existing methods applied to an unexplored system.
3.	The experimental setup is not clear. Details like training hyperparameters, ANN-SNN conversion parameters, and HW-level constraints of the neuromorphic chip that has been considered in the implementation.
4.	It is ok to keep anonymous the name and brand of the in-house neuromorphic chip on which the experiments have been conducted, but the experiments can also be conducted on publicly available neuromorphic chips, to ease the comparison with related works.
5.	For all the attacks, in particular for the universal attack, the stealthiness, i.e, how much the adversarial examples are noticed by the human eye, should be discussed.
6.	The results can be extended with experiments in which adversarial defenses, such as adversarial training or noise filter, are applied to improve the robustness against the attacks.


**Summary Of The Paper:**

In this paper, the adversarial attack algorithms have been adapted to be applied to event-based data for spiking neural networks implemented on neuromorphic hardware.

**Summary Of The Review:**

This paper tackles a very important problem with rigor and soundness. However, some key points need to be addressed (see the main review section).

---

> ### Author Response · Authors · 2021-11-22
> **Addressing reviewer 2Mf5's concerns**
>
> We would like to generally thank all reviewers for their respectful, clear, and fair reviews. We have addressed your concerns as follows:
>
> 1. We have now moved more details about the algorithms to the main text, and have removed the algorithm schematics, which were out of context and with poor descriptions of mathematical symbols. For detailed mathematical descriptions we refer to the original papers these were taken from, but, instead, we explain the changes we made in text, in the main paper. Section 2.1 has therefore been significantly expanded.
> 2. Applying methods that are suited for traditional networks with their continuous activations to spiking networks and DVS data is typically not a trivial task. In this case, for example, the discrete nature of the data meant we had to explore methods for updating the data without vanishing gradients. Additionally, we are the first who test the attacks against the network actually deployed on chip.
> 3. More detail about the neuromorphic chip will be included once the paper is de-anonymised and the chip can be described in detail. As mentioned in section 2.4, 8-bit weight precision results in quantization error on top of reduced temporal resolution during training. Regarding training parameters, we have added detail on the values of SNN thresholds, on the procedure of layer-wise normalisation, on the characteristics of the surrogate gradient function, and on the training of the ANN on NMNIST.
> For detailed information about the implementation, we refer the reader to the code attached in the supplementary material, which will be made publicly available.
> 4. We agree with this observation, however, this is the hardest to address, and we are forced to leave it for future work. Working with different neuromorphic chips requires considerable overhead in terms of boilerplate code and expertise, and there are usually differences in the exact neuron models and memory constraints, which usually require chip-specific solutions for modelling and training. For these reasons, comparing results of the same network or algorithm across multiple hardware brands is, to the best of our knowledge, not common in the neuromorphic community. Given the nascent state of the field, neuromorphic chips are typically not available for purchase and if research prototypes are available, they typically cost a multitude of what any latest-generation GPU would cost. The prohibitive access/cost combined with the inherent heterogeneity among neuromorphic hardware, with many different supported neuron models and configurations, make the comparison across different chips a very difficult task.
> 5. Regarding stealthiness, there is indeed a large difference between the offline attacks (Sparsefool or PGA) and the universal “patches” attack. For SparseFool, the attacks are often not noticeable — a measure of stealthiness is given by the number of spikes added or removed, which is extensively discussed in Table 1 and the lower-left graph of supplementary figure 1. Conversely, as the reviewer points out, the “patches” attacks end up adding large amounts of spikes in order to achieve the misclassification, and are therefore clearly noticeable to the human eye, as seen in Figure 3. However, these attacks have a different purpose: they show that attacking the network is in principle possible even without knowing the input in advance, while the stealthier attacks require prior knowledge of the input, and extensive computation which iteratively modifies it.  A patch attack could in principle be deployed in the real world by means of some flashing screen in front of the DVS camera, while this is not possible with non-universal attacks. Indeed, we must stress that the patches are also very well visible to the human eye also in the original patches paper (Brown et al. 2018), and this algorithm was never meant to be stealthy in this sense. Analogously, the attacks presented in Eykholt et al. 2018 are not “stealthy” as such (compared to a simple L2 attack), since they are made by physically adding distinctive stickers to real-world street signs, but they are in fact more dangerous. While the patch attack does not go all the way to a physically deployed attack, it already shows proof that online attacks are possible, which is a necessary condition.
> 6. By running additional experiments on the IBM Gestures dataset, we addressed the question whether the robustness of spiking CNNs to adversarial perturbations can be improved. For this, we resorted to TRADES (Zhang et al. 2019), a well-known method for increasing robustness to adversarial attacks. We tried several values of  the parameter that controls the trade-off between performance and robustness and show that with increasing emphasis on robustness, SparseFool needs a much larger number of added events in order to fool the network. However, the success rate remained high. We added these results to the supplementary material.

---

> > ### Comment · Reviewer_2Mf5 · 2021-11-29
> > **Response to authors**
> >
> > The effort made by the authors to address the reviewers' comments is appreciated. In light of the authors' responses and other reviewers' comments, I confirm my score.

---

### Decision · Program_Chairs · 2022-01-20

**Decision:**

Reject

**Comment:**

The manuscript investigates common adversarial attacks on event-based data for spiking neural networks.
They conclude that also in this setup adversarial attacks can strongly harm SNN performance.

Although the reviewers agree that the paper presents some solid results and is well written, there was also substantial criticism.

The main points were:
- It is not very clear how the usual attacks are applied to event-based data, and in general experimental setups are unclear.
- The methodological contribution of the paper seems limited.
- The novelty is limited, in particular Marchisio et al. 2021 investigates a very similar question and goes somewhat further. The author noted that their attacks are not deployed on neuromorphic hardware. A number of other important prior work is not discussed.
- The impact of adversarial defences was not considered.
- A more detailed comparison of event-based attacks to standard ANN attacks would be desired.

After the reviews, the authors have invested substantial efforts to improve the paper. These efforts were appreciated by the reviewers. In particular, the authors ran additional experiments using the defence method TRADES. The results showed that TRADES is effective, but the attack has still a large success.

In summary, the reviewers agree that this is a solid manuscript and an interesting direction, however, they see it finally slightly below acceptance threshold for ICLR.